# On the importance of the electric double layer structure in aqueous electrocatalysis

Seung-Jae Shin [1,6], Dong Hyun Kim[2,6], Geunsu Bae [2,6], Stefan Ringe [3,4], Hansol Choi [2], Hyung-Kyu Lim[5], Chang Hyuck Choi [2✉] & Hyungjun Kim [1✉]

To design electrochemical interfaces for efficient electric-chemical energy interconversion, it is critical to reveal the electric double layer (EDL) structure and relate it with electrochemical activity; nonetheless, this has been a long-standing challenge. Of particular, no molecular-level theories have fully explained the characteristic two peaks arising in the potential-dependence of the EDL capacitance, which is sensitively dependent on the EDL structure. We herein demonstrate that our first-principles-based molecular simulation reproduces the experimental capacitance peaks. The origin of two peaks emerging at anodic and cathodic potentials is unveiled to be an electrosorption of ions and a structural phase transition, respectively. We further find a cation complexation gradually modifies the EDL structure and the field strength, which linearly scales the carbon dioxide reduction activity. This study deciphers the complex structural response of the EDL and highlights its catalytic importance, which bridges the mechanistic gap between the EDL structure and electrocatalysis.

[1] Department of Chemistry, Korea Advanced Institute of Science and Technology, Daejeon 34141, Republic of Korea. [2] School of Materials Science and Engineering, Gwangju Institute of Science and Technology, Gwangju 61005, Republic of Korea. [3] Department of Energy Science and Engineering, Daegu Gyeongbuk Institute of Science and Technology, Daegu 42988, Republic of Korea. [4] Energy Science and Engineering Research Center, Daegu Gyeongbuk Institute of Science and Technology (DGIST), Daegu 42988, Republic of Korea. [5] Division of Chemical Engineering and Bioengineering, Kangwon National University, Chuncheon, Gangwon-do 24341, Republic of Korea. [6]These authors contributed equally: Seung-Jae Shin, Dong Hyun Kim, Geunsu Bae. ✉email: chchoi@gist.ac.kr; linus16@kaist.ac.kr

Electrocatalysis lies at the core of most modern technologies, such as, fuel cells, electrolyzers, and carbon dioxide recycling, for sustainable energy conversion. All such processes separate into half-cells in which electrochemistry happens under an electrochemical potential difference between the cathode and anode. The application of a potential difference leads to the formation of an electric double layer (EDL) at the interface of an electrode and liquid electrolyte. The EDL is one of the oldest and most fundamental concepts in electrochemistry[1,2]. As a recent example, the electrochemical carbon dioxide reduction reaction ($CO_2RR$) has been suggested to be controlled by the EDL structure[3–10].

Nonetheless, to date, the microscopic structure of the EDL has not been fully resolved not only because the EDL is spatially concealed between the two bulk phases of solid and liquid[11], but also because the electrochemical signals are highly convoluted by the complex, coupled EDL responses of the multiple components in the electrified interface[12]. Despite the recent successes based on X-ray absorption spectroscopy[13] and shell-isolated nanoparticle-enhanced Raman spectroscopy (SHINERS)[14], these spectroscopy-based investigations have been focused only to the explanation of the water orientations, and their quantitative association with electrocatalysis is yet to be established. However, a notable point of these studies is that the computational simulation has inevitably been employed; the simulated and experimental spectra have been matched with each other, based on which the structural details about the EDL have been obtained from the molecular simulations.

Instead of explaining the peaks from photon-based spectroscopy, we herein demonstrate that our molecular simulation accurately reproduces the characteristic peaks from an electrochemical impedance spectroscopy—the famous camel-shaped curve[15–19] of the capacitance in dilute aqueous electrolyte—without the requirement of empirical adjustment in the simulation. To reliably compute the differential capacitance, $C$, using its definition of $C = \partial\sigma/\partial E$ (where $\sigma$ is a surface charge density, and $E$ is an electrode potential), the sensitive changes in the potential must be captured that require an extremely fine sampling of the data points, which is practically impossible using a full ab initio approach[20]. Therefore, we utilize a multiscale approach, density functional theory in classical explicit solvents (DFT-CES), that combines a density functional description of the metal electrode with a classical molecular dynamic description of the electrolyte[21]. Most notably, the interfacial interaction of the DFT-CES is developed as based on the quantum−mechanical energetics, i.e., so-called first-principles based multiscale approach, that enables us to directly validate our simulation results through comparison with experimental results (for the simulation details and backgrounds, the reader may refer to our previous publications[21–23], and a summary is also presented in the Supplementary Notes 1–5).

## Results and discussion

With varying the $\sigma$ by changing the number of excess electrons in the Ag(111) electrode and excess ions ($Na^+$ or $F^-$) in the electrolyte, DFT-CES simulation of the interfacial system (Fig. 1a) predicts the change of the $E$, which is calculated using Trasatti's absolute electrode potential[24] (Supplementary Fig. 1 and Supplementary Fig. 2), yielding a $\sigma$–$E$ curve (Fig. 1b). Prior to evaluating the $C$, which is derivative of $\sigma$, we observe an unexpected feature in the $\sigma$–$E$ curve: an S-shaped region in the negative $\sigma$ region. Thermodynamically, the S-shaped profile is a consequence of a bistable free energy landscape (its microscopic origin will be discussed in the following section) by considering the thermodynamic relation of $dA = -SdT + Ed\sigma$ (where $A$ and $S$

are the interfacial Helmholtz free energy and entropy, respectively, and $T$ is the temperature). Thus, a Maxwell tie-line can be constructed along which two bistable states coexist in equilibrium (Supplementary Fig. 3). Such a phase coexistence line in the free energy curve yields a horizontal (or vertical) line in the $E$–$\sigma$ (or $\sigma$–$E$) curve that eventually causes the $C$–$E$ curve, a derivative of the $\sigma$–$E$ curve, to exhibit a capacitance peak.

Our theoretically predicted EDL capacitance curve is well matched with the curve corresponding to the experimental staircase potentiostatic electrochemical impedance spectroscopy (SPEIS) data measured for Ag(111) in a dilute 3 mM NaF electrolyte (Fig. 1c). In particular, the double-hump camel-shape of the capacitance curve is successfully reproduced at the peak potentials, comparable to that observed in the experiment within approximately 0.1 V. Both theoretical and experimental $C$–$E$ curves exhibit a minimum capacitance at the same $E$, that exactly corresponds to the point of zero charge (PZC) potential ($E_{PZC}$). In addition, the capacitance values are predicted to be approximately 20 $\mu F\ cm^{-2}$ near $E_{PZC}$ and in the highly polarized regions beyond the two humps on the curve; this is also consistent with the experimental results[17]. Theoretical peak behaviors are more exaggerated than those observed in the experiments and this can be attributed to the adiabatic potential change in theory. Indeed, the sharpness of the peaks increases as the potential sweep rate decreases in the experiments[15]. Therefore, we are now ready to elucidate the microscopic structural details of the EDL, which have been questioned but not fully resolved since the development of the early EDL theories in the 1900s[2]. In particular, we focus on what type of molecular structural response in the EDL is responsible for the two humps in the camel-shaped capacitance curves that have been measured from simple systems, such as the interfaces between planar metal electrodes and dilute aqueous electrolytes[25–34].

**Molecular origin of the camel-shape.** The key components of the EDL are water molecules, excess charges stored in the metal electrode, and ions in the electrolyte; all the local profiles of these along the $z$-direction are summarized to illustrate the complete structural details of the EDL in Fig. 1d. Hereafter, the $z$-directional distances are referenced relative to the center of the top-surface atoms of the metal electrode.

The local water density profile, $\rho_{wat}$, shows that two water layers are formed near the electrode at around $z = 3$ and 6 Å at all applied potentials, wherein the first layer is significantly adsorbed by the metal electrode (Supplementary Fig. 4). Therefore, it is reasonable to define the location of the inner Helmholtz layer (IHL) with respect to the location of this water adlayer[33]. Further, the ion charge density profile, $\rho_{ion}$, exhibits a peak at around $z = 5$ Å that is attributed to the solvated ions near the electrode, and it is used to define the location of the outer Helmholtz layer (OHL)[33]. In addition, a region beyond the OHL is a diffuse layer (DifL). The excess charge density profile of the electrode, $\rho_{electrode}$, is calculated by considering the difference between the electron charge density of the (dis)charged electrode and the uncharged electrode at the PZC. The value of $\rho_{electrode}$ indicates that electrons are added to or subtracted from the electron charge density tail in the profile of the metal surface, upon applying cathodic ($E < E_{PZC}$) or anodic ($E > E_{PZC}$) potential, respectively.

When the electrode is positively charged ($E > E_{PZC}$), the $F^-$ ions at the OHL are desolvated (Supplementary Fig. 5) and adsorbed on the electrode surface that increases the ion concentration at the IHL and decreases the ion concentration at the OHL (compare the change in the two peaks of $\rho_{ion}$ at $z = 2$ and 5 Å in the right panel of Fig. 1d). This specific adsorption of anions, proposed by Grahame[28] and demonstrated through

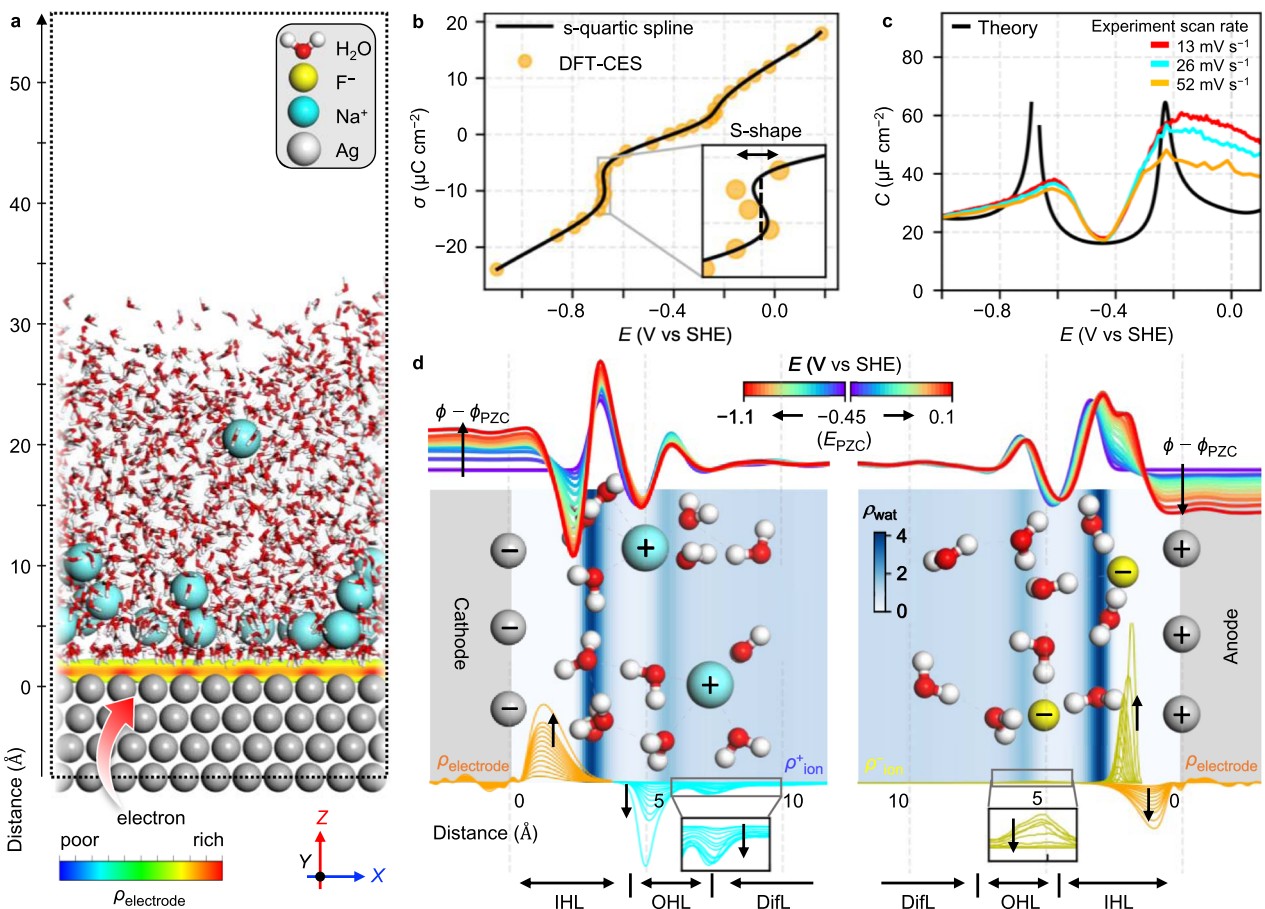

**Fig. 1 Molecular simulation of EDL charging curves and local profiles of constituents. a** A snapshot of the simulation system consisting of an Ag(111) electrode-electrolyte-vacuum interface. The excess charge density of the metal electrode, $\rho_{electrode}$, is shown as a color map that is screened by either excess Na$^+$ or F$^-$ ions in the electrolyte. **b** Surface charge density, $\sigma$, calculated as a function of the electrode potential, $E$ (vs standard hydrogen electrode (SHE)). A dashed vertical line is constructed along the coexisting line of two different charge states. The black solid line is smoothed quartic (s-quartic) spline fitting function of the DFT-CES data. **c** Comparison between the camel-shaped curve of the differential capacitance, $C$, versus $E$ with that of the experiments on Ag(111) electrode in a 3 mM NaF electrolyte. **d** Representative structures of hydrated ions near the electrodes. Local density profiles of water, $\rho_{wat}$, are shown as color maps in the background (unit: g cm$^{-3}$) that define the location of the IHL, OHL, and DifL. $\rho_{electrode}$, and the ion charge density profiles, $\rho_{ion}$, are also shown in the below row. Local electrostatic potential profiles, $\phi$, are shown in the upper row, as a function of $E$. The black arrows indicate the increase in negative or positive charging from the PZC. Source data are provided as a Source Data file.

various approaches[33,35], occurs because of their large dispersive attraction toward the electrode[36]. By mapping the double layer onto an effective two-plate capacitor, for an intuitive understanding, the specific adsorption is realized as a decrease in the charge-separation distance, $d$, of the ions from the charged electrode, resulting in an increase in the capacitance, by considering the relation, $C = \varepsilon_{eff}\varepsilon_0 A/d$ (where $A$ is the interfacial area, and $\varepsilon_0$ is the vacuum permittivity). The effective dielectric constant, $\varepsilon_{eff}$, quantifies the average field screening ability of the water dipoles in the EDL that is conceptually similar to the $\varepsilon_{eff}$ defined by Bockris, Devanathan, and Müller (BDM) in their seminal work on the BDM model[33]. The detailed quantification methods for $d$ and $\varepsilon_{eff}$ are described in the Supplementary Note 6.

Figure 2a shows that the anodic hump is an outcome of the capacitance increase, owing to the decrease in $d$ that is followed by the decrease in capacitance due to the decrease in and subsequent saturation of $\varepsilon_{eff}$. This trend is comparable to that in the previous dielectric saturation mechanism proposed by Booth[30], Conway et al.[31], Grahame[29], and Macdonald[32], developed after the Gouy[25]–Chapman[26]–Stern[27] theory. In addition, our simulation elucidates the molecular origin of dielectric saturation at a large anodic potential. When the anions are located at the OHL, they can accommodate the oxygen-

down (O-down) configuration of water molecules at the IHL and OHL (see upper left panels of Fig. 2b), mainly screening the interfacial electric field. However, when the anions are adsorbed on the electrode, the anions at the IHL stabilize the hydrogen-down (H-down) configuration of water molecules at the IHL and OHL, leading the dipole orientations to anti-screen the field at the interface. Consequently, the specific adsorption of the anions leads the interfacial water layers to exhibit no further macroscopic polarization when $\sigma > 5$ μC cm$^{-2}$ (upper right panel of Fig. 2b and c), causing a saturation behavior of $\varepsilon_{eff}$.

As widely presumed[37], when the electrode is negatively charged ($E < E_{PZC}$), the Na$^+$ ions are primarily accumulated at the OHL (Fig. 1d), keeping their first hydration shell intact (Supplementary Fig. 5) because of their low dispersive attraction toward the electrode[36], which is shown regardless of choice different water models (Supplementary Fig. 6). The absence of specific adsorption causes almost no change in $d$ (Fig. 2a), and therefore, the capacitance hump corresponding to the cathodic potential is primarily attributed to the change in $\varepsilon_{eff}$.

Upon the negative charging of the electrode, $\varepsilon_{eff}$ increases from 30 to 50, and subsequently decreases to 10, resulting in a peak at around $\sigma = -10$ μC cm$^{-2}$ (Fig. 2a). For the electrode less negatively charged than $-10$ μC cm$^{-2}$ ($-10 < \sigma < 0$ μC cm$^{-2}$),

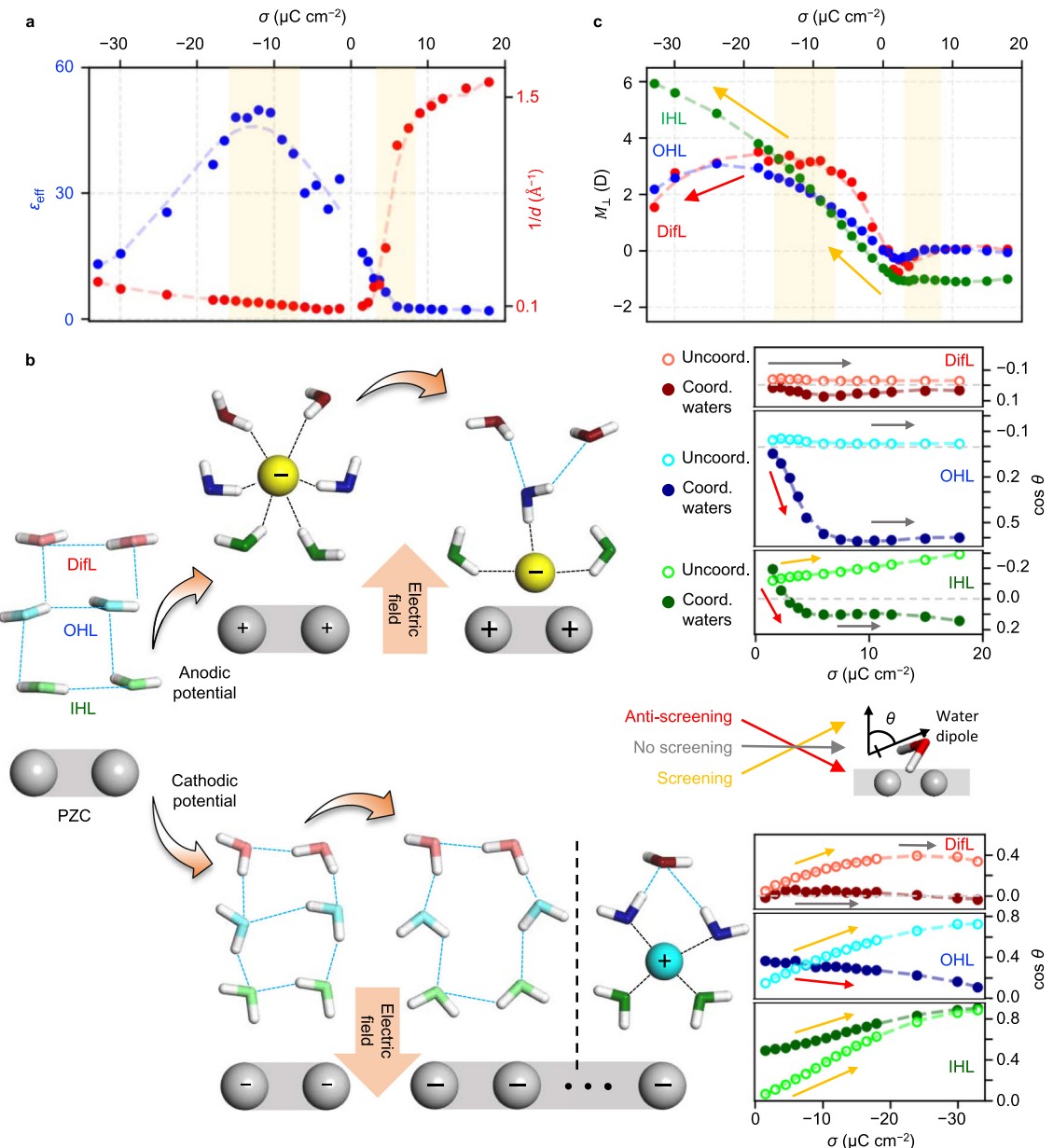

**Fig. 2 EDL structural responses upon anodic and cathodic charging. a** Reciprocal value of charge-separation distance, $1/d$, and effective dielectric constant, $\varepsilon_{eff}$, are shown as a function of the surface charge density, $\sigma$. The regions shaded pale-yellow correspond to the $\sigma$-range responsible for the humps in the differential capacitance curve. **b** Schematics illustrating the structural changes of water molecules and ions upon EDL charging from the PZC (left panels). Distinct orientational responses of the water dipoles are resolved depending on the layer at which water is located and also depending on whether the ion is coordinated (coord.) or not (uncoord.) (right panels), based on the average $\cos \theta$ ($\theta$ is the angle between the water dipole and the surface normal) that is a function of $\sigma$. Depending on whether the water dipole screens or anti-screens the field, the increasing or decreasing trend of $\cos \theta$ is labeled using the arrows with different colors. **c** Surface-normal macroscopic dipole moment, $M_\perp$, of different water layers is shown as a function of $\sigma$, where $M_\perp = \sum_{i \in \{IHL, OHL, DifL\}} m \cos \theta_i$ ($m$ is the water molecule dipole). Source data are provided as a Source Data file.

the water molecules at the interface rotate in a collective manner because of the hydrogen bond (HB) interaction via the following mechanism. As the electrode is negatively charged, the water@-IHL favors an H-down orientation (see lower left panels of Fig. 2b), in agreement with the findings of the previous in situ sum frequency generation[38] and SHINERS[14] studies. Such a molecular orientation offers an HB accepting O to the water@-OHL, leading the water@OHL to favor an H-down orientation and further promoting an H-down orientation of the water@DifL in a similar manner. Because of the cooperative behavior of water dipoles, a surface-normal macroscopic dipole of water layers, $M_\perp$, concurrently increases in all parts of the EDL (including IHL,

OHL, and DifL) for $-10 < \sigma < 0$ μC cm$^{-2}$ (Fig. 2c), increasing the magnitude of $\varepsilon_{eff}$. However, the negative charging of the electrode causes an accumulation of cations at the OHL that collapses the HB network formed across the IHL and OHL and therefore, impedes the cooperative rotation of the water dipoles (Supplementary Fig. 7). Furthermore, the cations accumulated at the OHL cause the nearby water at the OHL and DifL to have an O-down orientation that populates more anti-screening dipoles and thereby, decreases $M_\perp$ at the OHL and DifL (Fig. 2c), decreasing the magnitude of $\varepsilon_{eff}$ when $\sigma < -10$ μC cm$^{-2}$. Notably, this dielectric saturation mechanism is different from the previously suggested and broadly accepted speculations. Unlike

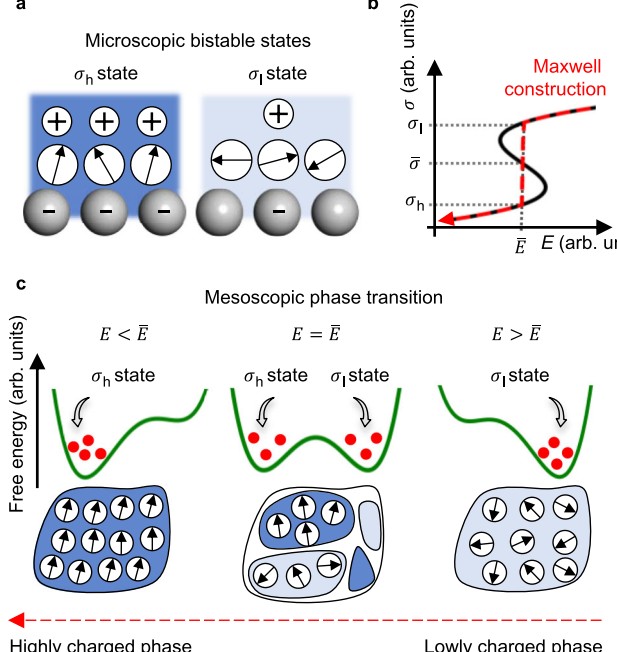

**a** Microscopic bistable states

$\sigma_h$ state   $\sigma_l$ state

**b** Maxwell construction

**c** Mesoscopic phase transition

$E < \bar{E}$   $E = \bar{E}$   $E > \bar{E}$

$\sigma_h$ state   $\sigma_h$ state   $\sigma_l$ state   $\sigma_l$ state

Highly charged phase   Lowly charged phase

**Fig. 3 EDL structural phase transition yields the cathodic hump. a** Higher charged state with more-aligned water dipoles (that is, large $|\sigma|$ and large $\varepsilon_{\text{eff}}$; $\sigma_h$ state) and lower charged state with less-aligned water dipoles (that is, small $|\sigma|$ and small $\varepsilon_{\text{eff}}$; $\sigma_l$ state) are bistable at the same cathodic potential, $E$. **b** Phase transition between bistable states, which is modeled using Landau-type theory, yields the S-shaped region, at which the Maxwell tie-line is constructed. $\bar{\sigma}$ is defined as $(\sigma_h + \sigma_l)/2$, and phase transition occurs at $E = \bar{E}$. **c** At the mesoscopically large interface, $\sigma_l$ state is more populated when $E > \bar{E}$, forming a lowly charged phase (right panels), and $\sigma_h$ state is more populated when $E < \bar{E}$, forming a highly charged phase (left panels). Negative-potential sweep induces a phase transition from the lowly charged phase to the highly charged phase via a phase coexistence at $E = \bar{E}$.

the previous assumption that the water at the IHL cannot rotate further and therefore shows no further orientation polarization beyond a critical field strength[30,31], the water at the IHL can indeed rotate its dipole further to screen the field (see lower right panel of Fig. 2b and c) because such a molecular orientation is accommodated by the cation at the OHL (see lower middle panel of Fig. 2b); nonetheless, the suppressed field-screening ability at the OHL and DifL leads to dielectric saturation.

**EDL structural phase transition**. We now demonstrate how the peak behavior of $\varepsilon_{\text{eff}}$ can result in the cathodic hump. For a two-plate capacitor model, the interfacial potential drop is proportional to $\sigma/\varepsilon_{\text{eff}}$. Consequently, when $\varepsilon_{\text{eff}}$ monotonically increases as $|\sigma|$ increases for $-10 < \sigma < 0$ $\mu$C cm$^{-2}$, as shown in Fig. 2a, two different surface charges, $\sigma_h$ and $\sigma_l$ (where $|\sigma_h| > |\sigma_l|$), can have the same interfacial potential drop. In other words, at the same cathodic potential of $E$, two states with different EDL structures characterized by $\sigma_h$ and $\sigma_l$ are bistable (Fig. 3a). Then, the Landau-type free energy density per interfacial area, $F$, is given as

$$F = \frac{\alpha}{2}(\sigma - \bar{\sigma})^2 + \frac{\beta}{4}(\sigma - \bar{\sigma})^4 - (E - \bar{E})(\sigma - \bar{\sigma}) \quad (1)$$

where $\alpha < 0$ for the bistable region, $\beta > 0$, and the two states have the same $F$ when $E = \bar{E}$, at which $\bar{\sigma} = (\sigma_h + \sigma_l)/2$. Using the Landau–Khalatnikov equation[39], we define the time variation of

EDL charging as,

$$R_s \frac{d\sigma}{dt} = -\frac{\partial F}{\partial \sigma} = E - \bar{E} - \alpha(\sigma - \bar{\sigma}) - \beta(\sigma - \bar{\sigma})^3 \quad (2)$$

where $R_s$ is the solution phase resistance. Using the equilibrium condition of $d\sigma/dt = 0$, we finally obtain the equilibrium $E$ as a cubic equation of $\sigma$,

$$E = \alpha(\sigma - \bar{\sigma}) + \beta(\sigma - \bar{\sigma})^3 + \bar{E} \quad (3)$$

that yields an S-shape in the $\sigma$–$E$ plane (black solid line in Fig. 3b). Therefore, the origin of the S-shaped region in Fig. 1b is attributed to the bistability of two different surface charge states in the cathodic potential range. Upon decreasing the potential from $E_{\text{PZC}}$ (by following the red dashed line in Fig. 3b), Landau-type theory predicts that an EDL structural phase transition will occur from the lowly charged phase ($E > \bar{E}$) to the highly charged phase ($E < \bar{E}$) at the mesoscopically large electrode–electrolyte interface; furthermore, phase coexistence occurs at $E = \bar{E}$ (Fig. 3c), where the cathodic peak emerges.

Notably, some theoretical models have predicted the emergence of a camel shape in the capacitance[34,40–42]; thus, it is useful to compare our approach to the previous model. One of the most recent and elaborate EDL models predicting the camel shape is the Kornyshev model[34,43–47], which is a lattice-gas model incorporating ion saturation behavior into the Gouy[25]–Chapman[26] theory, where water is coarse-grained as a dielectric. Although our mechanism for the cathodic hump indicates that the key to bistability is orientation polarization of the water molecular dipoles in the EDL, the Kornyshev model ascribes the emergence of the capacitance hump to ion saturation[34]. Thus, a concentrated electrolyte is essential to manifest a camel shape in the Kornyshev model, and in the dilute limit, the results of this model approach those of Gouy[25]–Chapman[26] model. Therefore, the Kornyshev model is suitable for explaining the camel-shaped capacitance measured in a dense Coulomb system such as an ionic-liquid electrolyte[34,43–48], whereas our mechanism explains the camel-shaped capacitance measured in a dilute aqueous electrolyte[15–19].

**EDL structure and electrocatalysis**. To modify the EDL structure near the cathodically polarized electrode, we utilize the strategy of the complexation of Na$^+$ with 15-Crown-5 (15C5). Through the DFT-CES simulation, we first identify that the 15C5 complexation prohibits the cation from being stably hydrated by the water at the IHL (Supplementary Fig. 8) that hinders the formation of a compact EDL structure and therefore increases $d$ approximately 1.5 times (Fig. 4a and Supplementary Fig. 9). This also increases the interfacial potential drop at the same $\sigma$ that shifts the S-shaped region to a more negative potential in the $\sigma$–$E$ plane (Supplementary Fig. 10). Consequently, our simulation predicts the negative potential shift of the cathodic hump in the $C$–$E$ curve that is also confirmed by our experiments (Fig. 4b).

The increase of $d$ leads to a weakened interfacial field when the same potential is applied at the interface (Fig. 4c). Therefore, the EDL structural modification through cation complexation provides an appropriate experimental platform for selectively controlling the field strength with maintaining the same electrode potential[49,50].

Recently, the mechanistic role of the local electric field in electrocatalytic reactions has been extensively discussed[5,7–9,49–52]. It is suggested that the rate of the electrochemical CO$_2$RR to carbon monoxide (CO), is limited by the CO$_2$ adsorption on the electrode surface, driven by the adsorbate dipole–field interaction[8]. From our CO$_2$RR experiments on Ag(111) with varying 15C5 concentration in the 100 mM NaHCO$_3$ electrolyte, we observe that the CO partial current density, $j_{\text{CO}}$, logarithmically decreases with an increasing 15C5 concentration (Fig. 4d

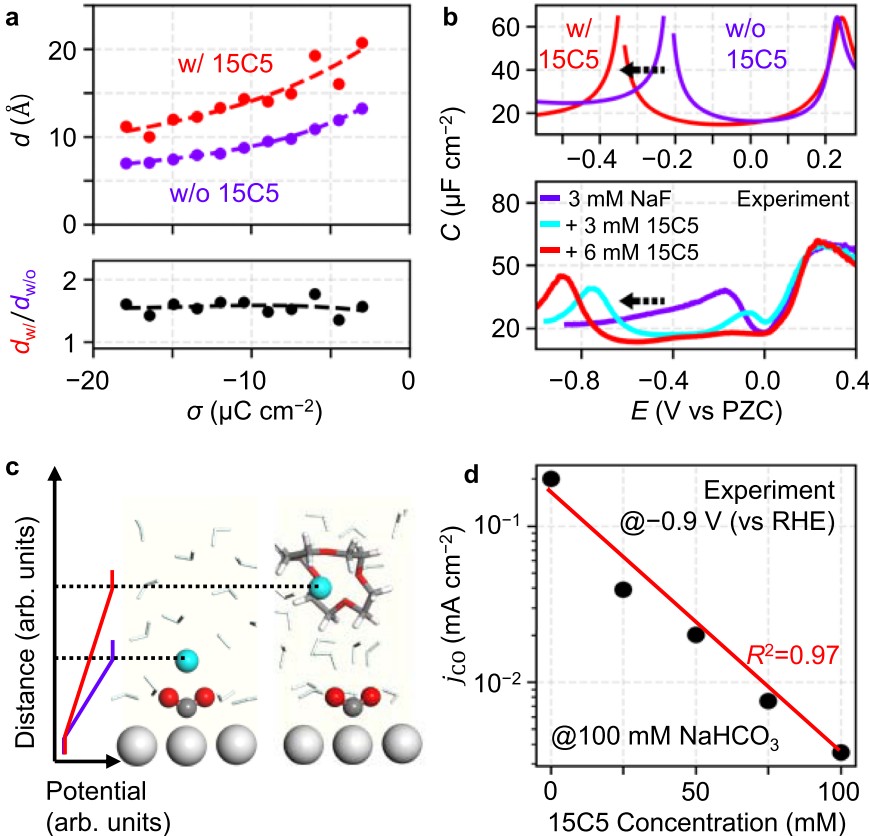

**Fig. 4 Structural change of EDL to modulate interfacial field strength and CO₂RR activity. a** Simulation-calculated charge-separation distance, $d$, is plotted as a function of the surface charge density, $\sigma$. The $d$ when Na⁺ is complexed with 15C5 ($d_{w/}$) is approximately two times larger than the $d$ when Na⁺ is uncomplexed ($d_{w/o}$). **b** A cathodic hump of differential capacitance, $C$, when Na⁺ is complexed with (w/) 15C5 shifts to a more negative potential than when Na⁺ is uncomplexed (w/o) in our simulation (upper panel), and experiments corroborate the negative-potential shift of the cathodic hump following the crown ether complexation (lower panel). **c** DFT-CES snapshots showing that uncomplexed Na⁺ develops a more direct interaction with the adsorbed CO₂ than 15C5-complexed Na⁺, forming a compact EDL structure with a stronger field. Also, the uncomplexed Na⁺ can easily make a direct coordination to the adsorbed CO₂. **d** A logarithmic dependence of CO partial current density, $j_{CO}$, on the 15C5 concentration that is experimentally measured for the Ag(111) electrode at −0.9 V (vs reversible hydrogen electrode (RHE)) in a CO₂-bubbled 100 mM NaHCO₃ electrolyte. Source data are provided as a Source Data file.

and Supplementary Fig. 11). We conceive that the macroscopically large EDL has a locally inhomogeneous structure comprising a compact part consisting of uncomplexed cations (with small $d$) and an uncondensed part consisting of 15C5-complexed cations (with large $d$); our experimental results indicate that the CO₂RR of the compact part of the structure, where the interfacial electric field is more intense, dominates the total activity, and therefore, a linear dependence of the log $j_{CO}$ on the ratio of the compact part that is considered to be proportional to the bulk 15C5 concentration, is exhibited. Not only the long-range dipole–field interaction, but also the short-range direct interaction of the cation with the adsorbate CO₂ has been highlighted recently[10,53]. Our DFT-CES simulation further revealed that the coordination number of Na⁺ to the adsorbed CO₂ decreases from 1.0 to 0.3 when the cation is complexed with 15C5 (Supplementary Fig. 12). Thus, the decrease in the CO₂RR activity can also be explained in terms of the decrease in the coordinating ability of a cation to the adsorbed CO₂. In both mechanistic possibilities, our work demonstrates the importance of identifying the EDL structure for controlling the electrocatalytic activity.

In summary, we have elucidated the complete structural details of the EDL based on a direct theory–experiment comparison of the EDL capacitance that is an electrochemical signal known to be sensitive to the EDL structure. This study demonstrates the ability to explore the detailed EDL structures based on a combination of the first-principles-based simulation and SPEIS experiments and

to further manipulate the electrocatalytic activity by tuning the EDL structure; this lays a foundation for establishing a link between the EDL structure and the electrocatalytic activity at a molecular level, that is a long-standing challenge in the electrochemistry.

## Methods

**DFT-CES simulations.** Our mean-field quantum mechanics/molecular mechanics (QM/MM) multiscale simulation, namely, DFT-CES[21], is implemented in our in-house code that combines the Quantum ESPRESSO[54] density functional theory simulation engine and LAMMPS[55] molecular dynamics simulation engine. Computational details can be found in the Supplementary Note 1.

**Electrochemical measurements.** Electrochemical measurements were conducted using an SP-150 potentiostat (Bio-Logic). An H-type electrochemical cell was fabricated using polyetheretherketone (PEEK) with a computer numerical control (CNC) milling machine (TinyCNC-SC, Tinyrobo). Each compartment of the cell had an opening area of $1.5 \times 1.5$ cm² on one side, where the electrolyte was separated by a Nafion 115 membrane (DuPont). A graphite rod and a saturated Ag/AgCl electrode (RE-1A, EC-Frontier) were used as the counter and reference electrodes, respectively. The reference electrode was doubly separated using a glass bridge tube to avoid halogen contamination[56]. A single crystalline Ag foil with a (111) orientation ($1 \times 1$ cm², 99.999%, MTI) was used as the working electrode, the surface of which was covered with Kapton tape, and an opening area of 0.16 cm² was ensured. The counter/reference and working electrodes were located in the different compartments of the electrochemical cell. Electrolytes were prepared by dissolving NaF (≥99%, Sigma-Aldrich) salts in ultrapure water (>18.2 MΩ, Arium® mini, Sartorius) with and without 15C5 (98%, Sigma-Aldrich).

Prior to each electrochemical measurement, the electrochemical cell was boiled in 0.5 M $H_2SO_4$ (98%, Daejung) and then ultrapure water for 2 h to clean the cell. The single-crystalline Ag electrode was chemically polished using the following procedure[57–59]. The Ag electrode was first immersed in a solution mixture of 0.3 M KCN (≥96%, Sigma-Aldrich) and $H_2O_2$ (29–32%, Alfa Aesar) with a volume ratio of 1.5/1 for 3 s, during which vigorous gas evolution occurred and thereafter, it was exposed to air for another 3 s. The Ag electrode was subsequently soaked in a 0.55 M KCN solution until gas evolution ceased, and it was thoroughly washed with ultrapure water. A highly reflective and homogenous surface was obtained after repeating the chemical polishing procedure 10 times. The Ag electrode surface was protected by a droplet of ultrapure water before it was transferred to the electrochemical cell. The differential capacitance was measured through SPEIS. The measurement was performed in a potential range from −1.3 to 0.2 V (vs SHE) with a frequency of 20 Hz and a potential amplitude of 10 mV in a deaerated electrolyte under Ar (5N) protection. The ohmic drop was compensated using a manual IR compensation (MIR, 85%) program during the SPEIS experiments.

The electrochemical $CO_2RR$ on the Ag(111) electrode was conducted in an H-type customized reactor consisting of separated compartments for the counter/reference and working electrodes, with a Nafion 115 membrane (DuPont). A 100 mM $NaHCO_3$ solution (≥99.7%, Sigma-Aldrich) with and without 15C5 was used as the electrolyte, in which $CO_2$ gas (5N) was continually bubbled at a flow rate of 20 sccm during the $CO_2RR$. A sequential chronoamperometry was conducted for 1 h at each potential in a range from −1.4 to −0.8 V (vs SHE). The reaction products, $H_2$ and CO, were monitored using an online gas chromatograph (GC; YL6500, YL Instrument) equipped with a thermal conductivity detector (TCD) and flame ionization detector (FID). A Carboxen-1000 column (12390-U, Supelco) was used for both TCD and FID, and Ar was used as the reference gas. All the potentials were compensated for IR loss.

## Data availability

All data is available in the main text or supplementary information. Source data are provided with this paper. The DFT-CES raw data generated in this study are provided in the Source Data file. Source data are provided with this paper.

## Code availability

The DFT-CES code has been deposited in the github database without accession code [https://github.com/SeungJay].

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

## Acknowledgements

This research was supported by the Samsung Science and Technology Foundation under Project Number SSTF-BA2101-08, and also by the National Research Foundation of Korea (NRF) grant funded by the Korea government (MSIT) (No. 2021R1A5A1030054). We also acknowledge the support by the Korea Institute of Science and Technology Information (KISTI) National Supercomputing Center with supercomputing resources including technical support (KSC-2020-CHA-0006).

## Author contributions

H.K. and C.H.C. supervised the project. H.K and H.-K.L. conceived the initial idea. S.-J.S. performed the DFT-CES simulation and analyzed the data. D.H.K. and G.B. performed the SPEIS experiments. S.R. analyzed the simulation results and contributed to manuscript editing. H.C. performed the $CO_2RR$ experiments. H.-K.L. contributed to development of algorithm of the DFT-CES simulation. All authors wrote and revised the manuscript.

## Competing interests

The authors declare no competing interests.
