## [Peer Review File · Nature Communications]

Title: On the importance of the electric double layer structure in aqueous electrocatalysisREVIEWER COMMENTS

Reviewer #1 (Remarks to the Author):

This work aims to demystify the structure of the electrochemical double layer using a combined computational and experimental approach. Through DFT calculations of a silver (111) surface the authors modeled the electrochemical double layer composed of Na⁺, F⁻, and water revealing the dynamic structure and transformations at the electrode/electrolyte interface upon polarization. The authors further performed experimental electrochemistry to understand the role that cations play in the EDL by complexing the sodium cations with a crown ether analog. Using the well characterized CO₂ reduction reaction as a platform to study the effects on complexed sodium ions the authors note a logarithmic decrease in current density in concert with increasing chelator concentration. The work presented herein provides a very systematic analysis of the electrochemical double layer structure

Notes:

- I recommend the authors consider revising the title to be more specific to the impact of the paper. This is a great paper, but it is hard to get excited by the title.

- The authors do not provide enough relevant literature as to the recent developments, both experimental and computational, in the field of interfacial electrocatalysis. It is therefore recommended that some of the works listed below be used to motivate the aims of this manuscript.

10.1039/C7CP06087D, doi.org/10.1063/1.5124878, doi.org/10.1039/C9EE01341E,
doi.org/10.1021/jacs.6b07612, doi.org/10.1021/jacs.7b06765, doi.org/10.1038/s41929-021-00655-5,
doi.org/10.1021/acs.jpcc.0c07004.

The authors provide a method for modelling the electrochemical double layer through DFT calculations revealing interesting molecular dynamics upon polarization. The experimental work contained herein supports computational findings by tying CO₂ electrocatalysis to the EDL structure. This work provides new insights and methods towards elucidating the double layer's structure/property reactions to electrocatalysis. The manuscript is detailed, organized, and insightful. I recommend this work for publication.

Reviewer #2 (Remarks to the Author): 
My review is in the attached file (because I needed to include graphics).

Reviewer #3 (Remarks to the Author):

The report is a study using mainly computational methods and a few experiments, to study the origin of the electric double layer. The system is a Ag(111) surface with water/Na⁺ or water/F⁻ solution. Several observed effects are very clearly reproduced and a credible molecular explanation is presented for the capacitance peaks. The paper is well-written and the conclusions are clearly presented. However, there are some things to address before the conclusions can be seen as verified. Since the conclusions fully rely on a single computational model which is non-standard and therefore not well tested, it is crucial that this model is benchmarked. Below are some specific points to address.

1. There are some questionable arguments in the molecular origin discussion. It is stated that the cause for F⁻ to adsorb on the surface while Na⁺ stays further away is due to the smaller hydration energy of anions. In the supporting information the hydration energy for F⁻ is presented as 115-120 kcal/mol, while Na⁺ has a value of 80-90 kcal/mol. This is precisely opposite to the argument on line 119. It is also stated that the dispersive energy is larger for F⁻ than for Na⁺, which is likely correct. This should be straight forward to estimate from the Uvdw term.
2. In the parametrization of the Buckingham potential for water, only the geometry with the oxygen adsorbed is probed. However, in the simulations at the cathode most water molecules point the hydrogen towards the surface. This geometry should also be probed, and should probably be tested with a charged Ag cluster.
3. It is not clear if the TIP3P Ag(111) interaction is balanced. I cannot find any benchmark of that. Especially when the surface is charged and the hydrogen points to the surface, the interaction in the presented model seems very strong so that the cations are even pushed out of the first layer. It could be correct, but it could as well be an artefact from a model that has a too strong interaction between H and Ag. Since TIP3P has an inflated charge distribution to compensate for the lack of anisotropy and other effects, it could lead to the formation of the silver-hydrogen bond formation at the cathode, which in turn seem to completely outcompete the silver cation interaction. I suggest that another water model is tested to see if the electrolyte structure is the same or if it is changed, to avoid the risk of an artefact due to a too simple water model.
4. The interpretation that the electric field difference is the determining factor for the difference in activity when crown-ether is added, could be correct but could as well be incorrect. Direct interaction between the oxygen atoms of -COO at the surface could also stabilize the formation of that adduct, and this interaction would also be limited by addition of crown-ether. There are some recent reports that discuss this phenomenon including Nature Catalysis 2021, 4, 654–662 and J. Phys. Chem. C 2020, 124, 41, 22479–22487.

Overall I believe that this report could provide very interesting and important insight on the catalyst-solvent interface under working conditions. There are some questions on the reliability of the method that needs to be addressed and some discussion that could be improved, but the key points of the paper are of high interest.

Comments to Reviewer. Thank you for your helpful comments, which are reproduced here in *italics*. Our responses are in **boldface**.

Reviewer: 1

Comments:

This work aims to demystify the structure of the electrochemical double layer using a combined computational and experimental approach. Through DFT calculations of a silver (111) surface the authors modeled the electrochemical double layer composed of Na⁺, F⁻, and water revealing the dynamic structure and transformations at the electrode/electrolyte interface upon polarization. The authors further performed experimental electrochemistry to understand the role that cations play in the EDL by complexing the sodium cations with a crown ether analog. Using the well characterized CO₂ reduction reaction as a platform to study the effects on complexed sodium ions the authors note a logarithmic decrease in current density in concert with increasing chelator concentration. The work presented herein provides a very systematic analysis of the electrochemical double layer structure

We would like to thank the reviewer for the favorable comments and recognition of the significance of our work.

Notes:

- I recommend the authors consider revising the title to be more specific to the impact of the paper. This is a great paper, but it is hard to get excited by the title.

We would like to thank the reviewer for the thoughtful suggestions regarding the title. Following the recommendations made by the reviewer, we changed the title from “Revealing the significance of the electric double layer structure for electrocatalysis” to “Electric double layer structure in aqueous electrolyte and its electrocatalytic importance” to reflect the impact of the paper more clearly.

- The authors do not provide enough relevant literature as to the recent developments, both experimental and computational, in the field of interfacial electrocatalysis. It is therefore recommended that some of the works listed below be used to motivate the aims of this manuscript.

10.1039/C7CP06087D, doi.org/10.1063/1.5124878, doi.org/10.1039/C9EE01341E, doi.org/10.1021/jacs.6b07612, doi.org/10.1021/jacs.7b06765, doi.org/10.1038/s41929-021-00655-5, doi.org/10.1021/acs.jpcc.0c07004.

We would like to thank the reviewer suggesting these additional references. We agree that all of the recommended references are important and have added corresponding citations on pages 3 and 12 of the revised manuscript.

The authors provide a method for modelling the electrochemical double layer through DFT calculations revealing interesting molecular dynamics upon polarization. The experimental work contained herein supports computational findings by tying CO₂ electrocatalysis to the EDL

structure. This work provides new insights and methods towards elucidating the double layer's structure/property reactions to electrocatalysis. The manuscript is detailed, organized, and insightful. I recommend this work for publication.

We appreciate the recognition of the reviewer regarding the importance of the EDL study, which will have a significant impact on the understanding of electrocatalysis. We also would like to thank the reviewer for recommending the publication of our work in *Nature Communications*.

Reviewer: 2

I have reviewed the manuscript titled “Revealing the significance of the electric double layer structure for electrocatalysis” by Shin et al. In brief, I think the manuscript has interesting scientific insight. However, the authors have entirely neglected previous literature and have made exaggerated claims about the nature of the double-hump capacitance curve (their main focus) as being entirely unexplained in the last 100 years. This lack of honest scholarship on their part, gives me serious pause about the paper. As I will discuss below, this problem has been addressed with many theoretical models in the past, and unlike what is claimed it is not the first time that it is explained. Their work is interesting, and perhaps should be published only and only after it is compared with existing theories and the exaggerations are toned down. I am not quite sure if this is the right journal for it, but that is the editor’s call. Similar papers have been published in the more specialized journal regularly (see examples below).

We appreciate the efforts made by the reviewer in critiquing our manuscript and providing us with valuable comments. We also would like to thank the reviewer for recognizing the scientific insights that the present work attempted to address. We fully acknowledge and admire the previous research works that have been devoted to electric double layers. Following the thoughtful suggestion provided by the reviewer, we avoided using unnecessary superlative expressions and revised the title to address the impact of our present work clearly.

We understand that the major concern of the reviewer is the lack of comparison of our results with previous theoretical works, such as the famous Kornyshev model. As the reviewer noted, the Kornyshev model successfully explains the emergence of camel-shaped differential capacitance [*Chem. Rev.* 114, 2978–3036 (2014)], which has been achieved by

including the ion saturation behavior in the Gouy-Chapman model. Consequently, the results of the Kornyshev model approach those of the Gouy-Chapman model at the dilute electrolyte limit [*Chem. Rev.* 114, 2978–3036 (2014)]; thus, it cannot be applied to dilute aqueous electrolyte systems, such as those of interest in this study, but more appropriately fits dense Coulomb systems such as ionic-liquid electrolyte systems. We would like to emphasize further the importance of aqueous electrochemistry in electrocatalysis. Electrocatalytic reactions in aqueous media offer an efficient means of converting small molecules (e.g., water, H₂, O₂, CO₂, NO_x, etc.) into other chemical species, coupled with renewable electricity storage and release.

To the best of our knowledge, there has been no full understanding of the molecular origin of the camel-shaped differential capacitance observed in dilute aqueous electrolyte systems, which we believe warrants the publication of our work in *Nature Communications*. We further believe that our fundamental understanding of the EDL structure in aqueous electrolytes provides insight into controlling the electrocatalytic reaction that is relevant to modern renewable energy technologies. A more detailed discussion is provided below.

Below I list my major critiques of this work:

1)- Unwarranted Superlatives and Exaggerated Claims:

The paper is replete with unwarranted superlatives. Here are some examples:

a)- In the abstract “Unprecedented structural phase transition”

Two issues here. First, the structural changes and phase transitions near the electrode as a function of potential is reasonably accepted and is not unprecedented. Second, just from a language point of view even if it were not the case, it is not the structural phase transition that is unprecedented, but rather its explanation.

I can't go through all of the literature discussing ionic structure and phase transitions near the surface. But as an example here is something from 25 years ago:

*Blum, L., Dale A. Huckaby, and M. Legault. "Phase transitions at electrode interfaces." *Electrochimica acta* 41.14 (1996): 2207-2227.*

*For a modern reference on this topic, please see below: Zhang, Yufan, et al. "Enforced freedom: electric-field-induced declustering of ionic-liquid ions in the electrical double layer." *Energy & Environmental Materials* 3.3 (2020): 414-420.*

This last paper sounds very similar to the claims of the paper under review. Similarly there are many like this in the literature.

We agree with the reviewer that the word “unprecedented” was unnecessary and thus have removed it from the manuscript. However, we would like to emphasize that the structural phase transition predicted from our cathodic-polarization simulation is different from the previously suggested ones. For example, the first paper that the reviewer mentioned [*Electrochim. Acta*, 41, 2207–2227, 1996] discussed a phase transition in the adsorbate layer of the ions, and the second paper by Kornyshev et al. [*Energy Environ. Mater.*, 3, 414–420, 2020] described a phase transition from bound cation–anion pairs to free ones in the moderate (or high) ion concentration regime, which cannot be related to the dilute aqueous

electrolyte case. Based on the comment provided by the reviewer that “I like the simulations, ionic structure change, and especially the Maxwell construction used by the authors. I think it is a decent explanation amongst many that already exist.”, we believe that the reviewer fully acknowledges that our work is distinctive from previous works but has a concern regarding our means of expression. We greatly appreciate the thoughtful suggestion and constructive comments provided by the reviewer and have revised our expressions accordingly.

b)- Another exaggerated claim is the following:

“While the EDL is one of the oldest concepts in electrochemistry¹, its significance in controlling electrochemical reactions has been recognized recently²⁻⁷.”

References 2-7 are all published in 2020-2021. Do the authors mean that no one in the last century ever recognized the importance of EDL for electrochemistry? Or they have something more specific in mind.

In continuation of our response in above, we deeply appreciate the constructive comments provided by the reviewer. We clarified the indicated expression as follows:

“The EDL is one of the oldest and most fundamental concepts in electrochemistry^{1,2}. As a recent example, the electrochemical carbon dioxide reduction reaction (CO₂RR) has been suggested to be controlled by the EDL structure³⁻¹⁰.”

c)- *This one is about spectroscopy of the EDL:*

“Nonetheless, to date, the atomic-level details of the EDL still remain unknown because not only the EDL is spatially concealed between the two bulk phases of solid and liquid, impeding spectroscopic measurements,...”

Have the authors not found any optical spectroscopic literature dedicated to the EDL structure that they can so confidently say the spectroscopic measurements are impeded at the EDL? There are hundreds, if not thousands of papers on spectroscopy of the electrode-electrolyte interface, including Raman, and IR vibrational spectroscopy dedicated to understanding ionic structure. A quote like this is inaccurate at best.

We agree that the original expressions could have been interpreted as ignoring the importance of spectroscopic dedications given in this field, although that was not our intention. Following the suggestion made by the reviewer, we have revised the indicated text as follows:

“Nonetheless, to date, the microscopic structure of the EDL has not been fully resolved not only because the EDL is spatially concealed between the two bulk phases of solid and liquid¹¹, but also because the electrochemical signals are highly convoluted by the complex, coupled EDL responses of the multiple components in the electrified interface¹².”

2)- *Lack of Comparison with Previous Models:*

One of the best known models of the double hump capacitance is described by Kornyshev's (and cited heavily in the literature).

Kornyshev, Alexei A. "Double-layer in ionic liquids: paradigm change?." (2007): 5545-5557.

*Fedorov, Maxim V., Nikolaj Georgi, and Alexei A. Kornyshev. "Double layer in ionic liquids: The nature of the camel shape of capacitance." *Electrochemistry Communications* 12.2 (2010): 296-299.*

*Chen, Ming, et al. "On the temperature dependence of the double layer capacitance of ionic liquids." *Journal of electroanalytical Chemistry* 819 (2018): 347-358.*

The figure to the right explains the camel shape of capacitance is from Kornyshev's 2007 paper mentioned above. If this is the case, then how come the authors claim the following in the paper: "We are now thus ready to address the century-year-long question, unresolved since the development of the early EDL theories in the 1900s: what type of molecular-scale change in the EDL structure is responsible for the two humps of the camel-shaped capacitance curve?²¹⁻³⁰."

Why is the above not referenced and contrasted with their new findings? This sounds very suspicious.

The authors could argue against the existing models if they wish, or show similarities and/or differences between the two approaches. However, the authors have circled around citing this well-known (and reasonably successful) model. Kornyshev is cited in passing only [ref 30], without any detailed comparison of the conceptual details. They have not even attempted to compare their

results with what is known. Avoiding Kornyshev's model is like avoiding an elephant in the room. Just to emphasize, this reviewer is not Kornyshev and has no association with him. I would be very much interested in understanding in what ways Kornyshev's lattice model and this work's phase transition ideas are similar or different.

Please note that it is not only Kornyshev who has models for the double hump capacitance. Here is an example from the more recent literature (and see the attached figure explaining the double hump capacitance:

*Cruz, Carolina, et al. "Electrical double layers close to ionic liquid–solvent demixing." *The Journal of Physical Chemistry C* 123.3 (2018): 1596-1601.*

First, the lack of comparison of our results with those of the Kornyshev model was an oversight on our part. As the reviewer noted below, the contribution of Kornyshev to the EDL field has been so great and his model is highly important in this field. However, we believe that this model (as well as many other models relying on a coarse-grained description of water as a continuum dielectric) has intrinsic limitations in resolving the microscopic origin of capacitance behavior measured in dilute electrolytes. First, the Kornyshev model is a lattice-gas model that does not include explicit treatment of water molecules. Although we do not criticize such a theoretical setting, as it also provides important insight with analytical convenience, this model cannot fully reflect the response of water molecules to the EDL field. The role of water molecules in the EDL becomes important for dilute, that is, water-rich electrolytes. Second, as mentioned previously, the essence of the Kornyshev model is the inclusion of ion saturation behavior in the Gouy-Chapman model, and the emergence of a camel shape is ascribed to ion saturation. Thus, a concentrated electrolyte is required to

manifest a camel shape, and in the dilute limit, the results of this model simply approach those of the Gouy-Chapman model, predicting a U-shaped capacitance curve. Therefore, the lack of explicit water molecules and the requirement of a finite ion concentration make the Kornyshev model suitable for explaining the camel-shaped capacitance measured in the ionic-liquid electrolyte system, rather than that measured in the dilute aqueous electrolyte system. In addition, owing to the lack of explicit water molecular structures in the model, the phase transition concept discussed based on the Kornyshev model is mostly related to the ion structure changes (e.g., from bound ion pairs to free ions [*Energy Environ. Mater.*, 3, 414–420, 2020]), whereas the phase transition predicted from our cathodic charging simulation has direct implications for the orientation response of the water dipoles in the EDL.

Following the comment made by the reviewer, we added the following paragraph to compare our results with those of the Kornyshev model on pages 11 and 12 of the revised manuscript:

“Notably, some theoretical models have predicted the emergence of a camel shape in the capacitance^{34,40–42}; thus, it is useful to compare our approach to the previous model. One of the most recent and elaborate EDL models predicting the camel shape is the Kornyshev model^{34,43–47}, which is a lattice-gas model incorporating ion saturation behavior into the Gouy²⁵–Chapman²⁶ theory, where water is coarse-grained as a dielectric. Although our mechanism for the cathodic hump indicates that the key to bistability is orientation polarization of the water molecular dipoles in the EDL, the Kornyshev model ascribes the emergence of the capacitance hump to ion saturation³⁴. Thus, a concentrated electrolyte is essential to manifest a camel shape in the Kornyshev model, and in the dilute limit, the results of this model approach those of Gouy²⁵–Chapman²⁶ model. Therefore, the Kornyshev model

is suitable for explaining the camel-shaped capacitance measured in a dense Coulomb system such as an ionic-liquid electrolyte^{34,43–48}, whereas our mechanism explains the camel-shaped capacitance measured in a dilute aqueous electrolyte^{15–19}.”

As discussed so far, our results are distinct from those of the previous models, which were mostly based on coarse-grained descriptions and/or intended to explain the camel-shape behavior of ionic-liquid systems. In this regard, to the best of our knowledge, no theory has provided molecular-level understanding of the origin of camel-shaped capacitance characteristically measured from a simple system, such as the interface between a planar metal electrode and dilute aqueous electrolyte. However, we agree with the thoughtful comment provided by the reviewer that some expressions, which sound exaggerated, needed to be toned down and clarified. Therefore, we changed the original text reading

“We are now thus ready to address the century-year-long question, unresolved since the development of the early EDL theories in the 1900s: what type of molecular-scale change in the EDL structure is responsible for the two humps of the camel-shaped capacitance curve?
21–30”

to

“Therefore, we are now ready to elucidate the microscopic structural details of the EDL, which have been questioned but not fully resolved since the development of the early EDL theories in the 1900s². In particular, we focus on what type of molecular structural response in the EDL is responsible for the two humps in the camel-shaped capacitance curves that

have been measured from simple systems, such as the interfaces between planar metal electrodes and dilute aqueous electrolytes²⁵⁻³⁴.”

I do like the explanations given the paper under review on the origin of the double hump capacitance. I like the simulations, ionic structure change, and especially the Maxwell construction used by the authors. I think it is a decent explanation amongst many that already exist.

We would like to thank the reviewer for recognizing the significance of our work. We sincerely appreciate the efforts made by the reviewer in critiquing our manuscript and providing constructive comments. Thanks to the thoughtful comments provided by the reviewer, our manuscript has been greatly improved and the distinct points of our study have been well clarified.

3)- *Relation to CO₂ reduction: The authors efforts in relating this work to CO₂ reduction is appreciated. However, it is done in passing and without much depth. Perhaps it can be done in a separate paper. The explanation in the manuscript is not in depth enough.*

We appreciate the reviewer for noting this issue. Indeed, after the submission of our manuscript, we became aware of the paper by Koper et al., suggesting the importance of short-range electrostatic interactions between cations and adsorbed CO₂ [*Nat. Catal.*, 4, 654–662 (2021)]. Thus, we further performed DFT-CES simulations to understand the interaction between the cation and adsorbed CO₂ in a bent form (*CO₂). Our DFT-CES simulation revealed that the cation can coordinate to the *CO₂ and that the coordinating ability of the cation to *CO₂ is weakened when the cation is chelated by the crown ether. Specifically, the coordination number of cation to *CO₂ decreases from 1.0 to 0.3 when the cation is chelated (Figure R1). This finding suggests that the linear dependence of the CO₂RR activity on the crown ether concentration can also be explained using a mechanism based on a direct cation–*CO₂ interaction.

Figure R1. Radial distribution function, $g_{Na-O}(r)$ and its integrated value, $\int 4\pi r^2 g_{Na-O}(r) dr$ as a function of a distance between Na^+ and O of adsorbed CO_2 . a, b, $g_{Na-O}(r)$ for uncomplexed Na^+ (a) and value of $\int 4\pi r^2 g_{Na-O}(r) dr$ (b). c, d, $g_{Na-O}(r)$ for 15C5-complexed Na^+ (c) and value of $\int 4\pi r^2 g_{Na-O}(r) dr$ (d).

However, as described in the original manuscript, the linear dependence of the CO_2RR activity on the crown ether concentration can also be explained in terms of the weakened electric field in the EDL. Thus, we believe that it is more appropriate to address both mechanistic possibilities in the present study than to close the discussion, which will inspire many other researchers to study the mechanism of CO_2RR further. Indeed, we are also

conducting a theoretical–experimental joint study to identify the mechanistic role of cations during CO₂RR, which will be presented in a future paper.

To address both possibilities based on short-range direct and long-range field-dipole interactions equally, we modified Figure 4c in the revised manuscript as shown in Figure R2.

Figure R2. DFT-CES snapshots showing that uncomplexed Na⁺ develops a more direct interaction with the adsorbed CO₂ than 15C5-complexed Na⁺, forming a compact EDL structure with a stronger field.

We also added Figure R1 into to the supporting information and appended below discussion to page 12 of the revised manuscript:

“Not only the long-range dipole–field interaction, but also the short-range direct interaction of the cation with the adsorbate CO₂ has been highlighted recently^{10,53}. Our DFT-CES simulation further revealed that the coordination number of Na⁺ to the adsorbed CO₂ decreases from 1.0 to 0.3 when the cation is complexed with 15C5 (Supplementary Fig. 12). Thus, the decrease in the CO₂RR activity can also be explained in terms of the decrease in the coordinating ability of a cation to the adsorbed CO₂. In both mechanistic possibilities, our work demonstrates the importance of identifying the EDL structure for controlling the electrocatalytic activity.”

Minor problems 1)- There are numerous English grammar and style problems in the manuscript. A couple of examples are below. In the abstract: “As based on first principles.” “that linearly scales the carbon dioxide reduction activity” 2)- The references section is split by the figures section. 3)- The unit Angstrom is missing in several places.

We properly revised the grammatical errors and carefully proof-read the manuscript.

Reviewer: 3

The report is a study using mainly computational methods and a few experiments, to study the origin of the electric double layer. The system is a Ag(111) surface with water/Na⁺ or water/F⁻ solution. Several observed effects are very clearly reproduced and a credible molecular explanation is presented for the capacitance peaks. The paper is well-written and the conclusions are clearly presented. However, there are some things to address before the conclusions can be seen as verified. Since the conclusions fully rely on a single computational model which is non-standard and therefore not well tested, it is crucial that this model is benchmarked. Below are some specific points to address.

We would like to thank the reviewer for the favorable comments and recognition of the significance of our present work. Following the valuable comments provided, we additionally performed an extensive benchmark study of our model, concluding that our findings are not parameter- or model-specific, but rather can be reproduced well in general. A more detailed discussion is provided below.

1. There are some questionable arguments in the molecular origin discussion. It is stated that the cause for F⁻ to adsorb on the surface while Na⁺ stays further away is due to the smaller hydration energy of anions. In the supporting information the hydration energy for F⁻ is presented as 115-120 kcal/mol, while Na⁺ has a value of 80-90 kcal/mol. This is precisely opposite to the argument on line 119. It is also stated that the dispersive energy is larger for F⁻ than for Na⁺, which is likely correct. This should be straight forward to estimate from the Uvdw term.

We agree with the reviewer that the adsorption behavior of F^- can be ascribed to the fact that its dispersive energy is larger than that of Na^+ . Indeed, the dispersion coefficients (i.e., C6 parameters) and F^- and Na^+ are $1012 \text{ kcal mol}^{-1} \text{ \AA}^6$ and $21 \text{ kcal mol}^{-1} \text{ \AA}^6$, respectively, as determined from real-time time-dependent density functional theory (RT-TDDFT) calculations [*J. Chem. Theory Comput.* 12, 3603–3613 (2016)], which were employed in our simulations. Thus, we changed the original text reading

“This specific adsorption of anions, proposed by Grahame²⁴ and demonstrated through various approaches^{29,31}, occurs because of the relatively small hydration free energy of anions³² and their large dispersive attraction toward the electrode³³.”

to

“This specific adsorption of anions, proposed by Grahame²⁸ and demonstrated through various approaches^{33,35}, occurs because of their large dispersive attraction toward the electrode³⁶.”

2. In the parametrization of the Buckingham potential for water, only the geometry with the oxygen adsorbed is probed. However, in the simulations at the cathode most water molecules point the hydrogen towards the surface. This geometry should also be probed, and should probably be tested with a charged Ag cluster.

We would like to thank the reviewer for validating our FF parameters. To respond to the question posed by the reviewer, we refitted our interfacial FF parameters to reproduce both

the O- and H-head interactions on the Ag surface. The original FF parameters only include Ag–O_{water} interactions, but new FF parameters include both Ag–O_{water} and Ag–H_{water} interactions using Buckingham potentials (see Table R1), which reproduce the nonlocal vdW-corrected DFT binding energy curves of water to the Ag(111) surface for both O- and H-head orientations (Figure R3).

		A (kcal mol ⁻¹)	R (Å)	C_6 (kcal mol ⁻¹ Å ⁶)
Original FF parameters	Ag–H _{water}	0	1.0	0
	Ag–O _{water}	28888	0.328	2009
New FF parameters	Ag–H _{water}	3389	0.343	355
	Ag–O _{water}	17427	0.348	1739

Table R1. Original and new force-field (FF) parameters for the Buckingham potential.

Figure R3. DFT-CES benchmark versus QM-level vdW-corrected DFT calculation results. a, b, Binding energy curves for the O-head geometry: binding energy curves (a) and O-head geometry (b). c, d, Binding energy curves for the H-head geometry; binding energy curves (c) and H-head geometry (d).

Using the new FF parameters, we re-calculated the charging curve (Figure R4a) and differential capacitance curve (Figure R4b). The newly obtained curves show the same characteristic features as the previously obtained curves using the original FF parameters, despite small quantitative changes in the potential values. In addition, the newly obtained

differential capacitance curve more closely matches the experimental curve in terms of the peak positions and capacitance at the potential at the point of zero charge (E_{PZC}). Thus, we updated all the data in the revised manuscript using the simulation results based on the new FF parameters. The key findings and conclusions remain the same despite some minor quantitative updates.

Figure R4. Comparison of DFT-CES results using original and new FFs. **a**, Surface charge density, σ , versus electrode potential, E . **b**, Differential capacitance, C , versus E .

As requested by the reviewer, we further validated our new FF parameters using a charged Ag cluster. Utilizing neutral and -1 charged Ag hexamer clusters, we calculated the QM-level binding energy curves of water molecules for O- and H-head orientations. Because of the lack of a nonlocal vdW-corrected functional of vdW-*df2*, which was used for slab calculations, in the non-periodic DFT codes, we employed the D3 vdW-correction method of

Grimme coupled with the HSE06 functional. As shown in Figure R5, we found that the DFT-CES based on the new FF parameters successfully reproduced the binding energy curves of water molecules to neutral and charged Ag hexamer clusters, both for O- and H-head orientations.

Figure R5. DFT-CES benchmark versus QM-level vdW-corrected DFT calculation results. a, b, c Binding energy curves of water to neutral Ag hexamer (a) and to -1 charged Ag hexamer (b) for O-head geometry (c). **d, e, f**, Binding energy curves of water to neutral Ag hexamer (d) and to -1 charged Ag hexamer (e) for H-head geometry (f).

3. It is not clear if the TIP3P Ag(111) interaction is balanced. I cannot find any benchmark of that. Especially when the surface is charged and the hydrogen points to the surface, the interaction in the presented model seems very strong so that the cations are even pushed out of the first layer. It could be correct, but it could as well be an artefact from a model that has a too strong interaction between H and Ag. Since TIP3P has an inflated charge distribution to compensate for the lack of anisotropy and other effects, it could lead to the formation of the silver-hydrogen bond formation at the cathode, which in turn seem to completely outcompete the silver cation interaction. I suggest that another water model is tested to see if the electrolyte structure is the same or if it is changed, to avoid the risk of an artefact due to a too simple water model.

We firstly note that the original FF parameters predicted nearly the same binding energy of water to the Ag surface for the H-head configuration but that the separation distance was slightly underestimated (Figure R3). Thus, very strong binding of water to the surface is unlikely to occur even when using the original FF parameters.

By using the new FF parameters, which were fitted to reproduce the Ag-H and Ag-O interactions equally, we also found that the first-coordination shell of the cation remained intact even for the highly charged Ag surface case, in agreement with the results obtained using the previous FF parameters (Figure R6).

Figure R6. DFT-CES simulation results showing the solvation structure of cations and the charge-separation distance, d , when using the original and new FFs. a, b, Coordination number (CN) of water to the cation (a) and d versus surface charge density, σ . (b) when using the original FF parameters. c, d, CN of water to the cation (c) and d versus σ (d) when using the new FF parameters.

To test any possible artefacts due to the use of a simple TIP3P-EW water model, as requested by the reviewer, we employed the TIP4P-EW water model and performed additional calculations. Even after changing the water model from TIP3P-EW to TIP4P-EW, we found

that the first-coordination shell of the cation remained intact even for the highly charged Ag surface (Figure R7).

Figure R7. DFT-CES simulation results showing the solvation structure of cations and the charge-separation distance, d , when using the TIP3P-EW and TIP4P-EW water models. a, b, Coordination number (CN) of water to the cation (a) and d versus surface charge density, σ , (b) when using the TIP3P-EW model (violet) and TIP4P-EW model (red).

We thus concluded that our finding is not an artefact caused by a specific choice of model parameters, but rather is physically sound.

4. The interpretation that the electric field difference is the determining factor for the difference in activity when crown-ether is added, could be correct but could as well be incorrect. Direct interaction between the oxygen atoms of $-COO$ at the surface could also stabilize the formation of that adduct, and this interaction would also be limited by addition of crown-ether. There are some recent reports that discuss this phenomenon including *Nature Catalysis* 2021, 4, 654–662 and *J. Phys. Chem. C* 2020, 124, 41, 22479–22487.

We appreciate the reviewer for noting this issue. Indeed, after submitting our manuscript, we became aware of the paper by Koper et al., which suggests the importance of short-range electrostatic interactions between cations and adsorbed CO_2 [*Nat. Catal.*, 4, 654–662 (2021)]. Thus, we further performed DFT-CES simulations to understand the interaction between the cation and adsorbed CO_2 in a bent form ($*CO_2$). Our DFT-CES simulation revealed that the cation can coordinate to the $*CO_2$ and that the coordinating ability of the cation to $*CO_2$ is weakened when the cation is chelated by the crown ether; the coordination number of cation to $*CO_2$ decreases from 1.0 to 0.3 when the cation is chelated (Figure R8). This finding suggests that the linear dependence of the CO_2RR activity on the crown ether concentration can also be explained using a mechanism based on a direct cation– $*CO_2$ interaction.

Figure R8. Radial distribution function, $g_{Na-O}(r)$ and its integrated value, $\int 4\pi r^2 g_{Na-O}(r) dr$ as a function of a distance between Na^+ and O of adsorbed CO_2 . a, b, $g_{Na-O}(r)$ for uncomplexed Na^+ (a) and value of $\int 4\pi r^2 g_{Na-O}(r) dr$ (b). c, d, $g_{Na-O}(r)$ for 15C5-complexed Na^+ (c) and value of $\int 4\pi r^2 g_{Na-O}(r) dr$ (d).

However, as described in the original manuscript, the linear dependence of the CO_2RR activity on the crown ether concentration can also be explained in terms of the weakened electric field in the EDL. Thus, we believe that it is more appropriate to address both mechanistic possibilities in the present study than to close the discussion, which will inspire many other researchers to study the mechanism of CO_2RR further. Indeed, we are also

conducting a theoretical–experimental joint study to identify the mechanistic role of cations during CO₂RR, which will be presented in a future paper.

To discuss both possibilities based on short-range direct and long-range field-dipole interactions equally, we modified Figure 4c in the revised manuscript as shown in Figure R9.

Figure R9. DFT-CES snapshots showing that uncomplexed Na⁺ develops a more direct interaction with the adsorbed CO₂ than 15C5-complexed Na⁺, forming a compact EDL structure with a stronger field.

“Not only the long-range dipole–field interaction, but also the short-range direct interaction of the cation with the adsorbate CO₂ has been highlighted recently^{10,53}. Our DFT-CES simulation also revealed that the coordination number of Na⁺ to the adsorbed CO₂ decreased from 1.0 to 0.3 when the cation was complexed with 15C5 (Supplementary Fig. 12). Thus, the decrease in the CO₂RR activity can also be explained in terms of the decrease in the

coordinating ability of a cation to the adsorbed CO₂. In both mechanistic possibilities, our work demonstrates the importance of identifying the EDL structure for controlling the electrocatalytic activity.”

Overall I believe that this report could provide very interesting and important insight on the catalyst-solvent interface under working conditions. There are some questions on the reliability of the method that needs to be addressed and some discussion that could be improved, but the key points of the paper are of high interest.

We would like to thank the reviewer for recognizing the significance of our work. We sincerely appreciate the efforts made by the reviewer in critiquing our manuscript and providing constructive comments. Thanks to the thoughtful comments provided by the reviewer, our manuscript has been greatly improved and clarified.

REVIEWERS' COMMENTS

Reviewer #2 (Remarks to the Author):

The authors have addressed most of my major concerns. At this point, their results merits publishing and being shared with the community.

Reviewer #3 (Remarks to the Author):

I find the changes and replies to the comments I made (reviewer 3) are very satisfactory. The newly developed model has a much better balance between the H-bound and O-bound conformations, and appears to give even better agreement with the experimental results. The explanation of the F- and Na+ interaction with the surface was aslo changed to one that made more sense from the data. From my point of view the manuscript is now suitable for publications.